# Genome-Wide Identification and Characterization of the YTH Domain-Containing RNA-Binding Protein Family in *Liriodendron chinense*

**DOI:** 10.3390/ijms242015189

**Published:** 2023-10-14

**Authors:** Sheng Yao, Jingjing Zhang, Xiang Cheng, Dengbao Wang, Wenya Yu, Kongshu Ji, Qiong Yu

**Affiliations:** 1State Key Laboratory of Tree Genetics and Breeding, Nanjing Forestry University, Nanjing 210037, China; yaosheng0817@163.com (S.Y.); jjzhang@njfu.edu.cn (J.Z.); chengxiang@njfu.edu.cn (X.C.); dbw@njfu.edu.cn (D.W.); ya163163@163.com (W.Y.); 2Key Open Laboratory of Forest Genetics and Gene Engineering of National Forestry & Grassland, Nanjing Forestry University, Nanjing 210037, China; 3Co-Innovation Center for Sustainable Forestry in Southern China, Nanjing Forestry University, Nanjing 210037, China

**Keywords:** RNA methylation, m^6^A, *Liriodendron chinense*, YTH domain, expression profiling

## Abstract

*N^6^*-methyladenosine (m^6^A) is becoming one of the most important RNA modifications in plant growth and development, including defense, cell differentiation, and secondary metabolism. YT521-B homology (YTH) domain-containing RNA-binding proteins, identified as m^6^A readers in epitranscriptomics, could affect the fate of m^6^A-containing RNA by recognizing and binding the m^6^A site. Therefore, the identification and study of the YTH gene family in *Liriodendron chinense* (*L. chinense*) can provide a molecular basis for the study of the role of m^6^A in *L. chinense*, but studies on the *YTH* gene in *L. chinense* have not been reported. We identified nine putative *YTH* gene models in the *L. chinense* genome, which can be divided into DF subgroups and DC subgroups. Domain sequence analysis showed that the LcYTH protein had high sequence conservation. A LcYTH aromatic cage bag is composed of tryptophan and tryptophan (WWW). PrLDs were found in the protein results of YTH, suggesting that these genes may be involved in the process of liquid–liquid phase separation. *LcYTH* genes have different tissue expression patterns, but the expression of *LcYTHDF2* is absolutely dominant in all tissues. In addition, the expression of the *LcYTH* genes is changed in response to ABA and MeJA. In this study, We identified and analyzed the expression pattern of *LcYTH* genes. Our results laid a foundation for further study of the function of the *LcYTH* gene and further genetic and functional analyses of m^6^A RNA modification in forest trees.

## 1. Introduction

*N^6^*-methyladenosine regulates RNA metabolism and gene expression, thereby controlling the reproduction, growth, and development in mammals and plants at the post-transcriptional level [1,2]. Current sequencing results have revealed that m^6^A methylation exhibits distinct patterns between plants and animals, leading to diverse functional roles in these two kingdoms. The diverse patterns indicate the diverse functional roles of m^6^A methylation in plants and animals [3,4,5]. However, the current research on m^6^A mainly focuses on humans and mammals, with limited studies on plants. Therefore, the aim of the present study was to investigate a plant m^6^A-modification system. The m^6^A-modification system consists of methyltransferases (writers) which are responsible for adding methyl groups, demethylases (erasers) that remove methyl groups, and recognition proteins (readers) that bind m^6^A sites to determine the fate of RNA [6,7,8,9,10,11].

As the important component in the m^6^A-modification system, the readers play vital roles in various biological processes, including pre-mRNA splicing, mRNA degradation, translation efficiency, and translation initiation specifically via recognizing and binding to the m^6^A sites on RNA molecules [12,13,14,15]. Readers were initially discovered in mammalian cells and are predominantly composed of a conserved YTH (YT521-B homology) domain, known as YTHDC1, YTHDC2, YTHDF1, YTHDF2, and YTHDF3 in humans [16]. The YTH domain, which is highly conserved in both plants and animals, contains crucial tryptophan residues that are essential for m^6^A binding [17,18,19]. Therefore, the proteins that possess these YTH domains were predicted to have the potential to interact with m^6^A sites in plants.

Previous studies on YTH proteins have predominantly focused on animals. With the booming of m^6^A in plant research, several studies on YTH protein in plants have been reported [20]. According to the current research, the plant YTH proteins can be classified into two subgroups, namely DF and DC [20]. Among these, the plant DF proteins can be further categorized into three subgroups, DFA, DFB, and DFC. However, in certain species that are considered outgroups for the angiosperms, only a single DF version is present, while chlorophyte species lack DF proteins altogether [19]. It has been observed that the aspartic acid (position 401 in human protein) of the DFB subgroup Helix α1 is replaced by asparagine [21]. This substitution of amino acid in the DFB subgroup of plant YTH proteins may significantly increase its affinity for m^6^A compared to proteins from the other two subgroups [21]. Additionally, apart from the highly conserved YTH domain, the N-terminal region (around 50 base pairs) of most plant DF proteins has a specific bias towards tyrosine, proline, and glutamine (YPQ-rich) [10]. This bias is likely to facilitate the aggregation of *HsYTHDF2* in processing bodies and promote protein-protein interactions [10]. The phylogenetic tree of the Viridiplantae DC group also revealed two subgroups, DCA and DCB [19]. In addition to the YTH domain, plant DCA proteins have three highly conserved N-terminal zinc fingers [19]. Surprisingly, while DCA and DCB proteins are present in all species that are outgroups to the angiosperms and in all dicotyledon species, no DCB protein was found in the monocotyledon lineage, suggesting that this version of the YTH motif was lost in the common ancestor of monocotyledons [19]. In *Arabidopsis thaliana* (*A. thaliana*), 13 proteins containing YTH domains were identified and named the EVOLUTIONARILY CONSERVED C-Terminal REGION (ECT) [22]. *ECT2*, the first m^6^A reader identified in *A. thaliana* [19,23], has been reported to be able to not only recognize the traditional m^6^A sites RRACH (R = A/G; H = A/C/U) but to also bind a plant-specific m^6^A conserved sequence URUAY (R = G > A, Y = U > A; A, in which UGUAY sequences accounted for more than 90%) [23]. Through binding this m^6^A site, *ECT2* can control trichome morphology by facilitating m^6^A-containing RNA stability in *A. thaliana*. Further investigations showed that *ECT3*, *ECT4*, and *ECT2* have complementary functions and all serve as m^6^A readers [24,25]. Additionally, this research has demonstrated that the recognition of m^6^A sites by *CPSF30-L* affects the process of liquid–liquid phase separation (LLPS) [17]. Studies on various plant species have revealed that YTH proteins play crucial roles in numerous plant processes such as embryogenesis, flowering transition, root development, stem cell fate determination, circadian rhythm, leaf morphology, leaf coat development, nitrate signaling, fruit maturation, gametophyte development, phytohormone and stresses responses [20]. Consequently, it is necessary to investigate the YTH gene in plants to further facilitate the understanding of the regulatory mechanisms of m^6^A on mRNA processing and metabolism in plants as well as its impact on plant growth and development.

*Liriodendron chinense* (*L. chinense*), a tree species of the Magnoliaceae family, an ancient relict plant, holds significant importance as it has been listed as a rare and endangered plant species in China. As one of the most precious tree species in the world, it not only has high ecological and landscape utilization value but also has great economic value serving as the source of precious wood. However, no studies on YTH protein have been reported on such a precious tree. This study comprehensively identified and characterized the *LcYTH* gene in *L. chinense* to gain deeper insights into the functions of *YTH* genes in the development and stress resistance of *L. chinense*.

## 2. Results

### 2.1. Genome-Wide Identification and Phylogenetic Analysis of L. chinense YTH Genes

We performed local BLASTP and HMMER searches, resulting in the identification of nine putative protein sequences (Figure 1A; Table 1 and Appendix A). Phylogenetic tree analysis showed that there were eight *YTH* genes belonging to the DF subgroup and one *YTH* gene belonging to the DC subgroup. We further constructed an evolutionary tree specifically for the DF genes. Phylogenetic analysis of the DF group showed the presence of LcYTHDF protein members in all three subgroups (Figure 1B). However, only one DC protein, LcCPSF30-L, was identified, which belongs to the DCA subgroup (Figure 1C).

### 2.2. Sequence Analysis of L. chinense YTH Genes

To investigate the characteristics of the genes, various parameters were analyzed, including their chromosomal location, amino acid length, molecular weight, and isoelectric point (pI) (Table 1 and Appendix A). Through the analysis of the chromosome distribution based on the *L. chinense* genome, nine *YTH* genes with genome annotation were found to be located on nine different chromosomes in *L. chinense* (Figure 2; Table 1). According to the analysis of gene features, the amino acid length of 9 LcYTH proteins varied from 427 (LcYTHDC6) to 744 (LcYTHDF2) amino acids, with molecular weights between 48.2 (LcYTHDC6) and 80.4 (LcYTHDF2) kDa. The predicted pI values ranged from 5.31 (LcYTHDF4) to 6.56 (LcYTHDF9).

### 2.3. Duplication Events of LcYTH Genes

To gain insights into the evolution of *LcYTH* genes, we investigated genome duplication events in this gene family. We utilized many gene families in plants that occur as a result of tandem or segment duplications. We used MCScanX in the TBtools software (v1.108 Chengjie Chen, Guangzhou China) to analyze the synteny relationship between *LcYTH* genes. As a result, we identified a total of two pairs of four genes with segmental duplication. The data were visualized using Circos in the TBtools software (v1.108 Chengjie Chen, Guangzhou, China) (Figure 3). To further understand the evolutionary pressures exerted on duplicated homologous genes of *LcYTH*, we measured the Ka/Ks nucleotide substitution ratios of collinear genes (Appendix A). The analysis of *LcYTH* genes revealed that their Ka/Ks << 1, indicating that the duplicated *LcYTH* genes have undergone a purifying selection during their evolutionary history. This suggests that the duplicated genes have been retained and selected for their functional relevance, likely to adapt to external changes.

### 2.4. Gene Structure, Conserved Domain, and Motif Analysis of LcYTH Genes

Different subfamilies within the LcYTH gene family exhibit distinct gene structure patterns. Most members within each subgroup of the LcYTH gene family share similar gene structure and gene length (Figure 4A,B). It was found that all LcYTH genes contain introns, with the majority of genes having between five to eight introns. However, there were also certain differences in gene structure among members of the same subfamily: for example, LcYTHDF5 possesses ultra-long introns (>8 kb). This diversity in gene structure suggests an evolutionary trend towards functional diversity among the LcYTH gene family.

Domain analysis showed that the YTH domains of the DF subgroup were located at the C-terminus, and the DC subgroup was located in the middle of the gene (Figure 4C). To further analyze the structural diversity in the LcYTH proteins, the conserved motifs in LcYTH proteins were identified using the online tool MEME (http://meme-suite.org/tools/meme (accessed on 4 March 2023)) (Appendix A). The analysis of conserved motifs revealed that all LcYTH proteins contain motifs 1, 2, 3, and 6. Additionally, motifs 4, 5, 7, 8, 9, and 10 were found exclusively in the DF subgroup. Motif 9 was only found in the DFA and DFB subgroups, while motif 10 was only found in the DFA subgroup (Figure 4D). This suggests that closely related genes identified through phylogenetic analysis also share similar motif types and distribution patterns. The YTH domain in the LcYTH proteins is composed of three conserved motifs (1, 2, and 3). Notably, the aromatic cage in the YTH domain of LcYTHs is composed of tryptophan residues (WWW) (Figure 4E; Appendix A). In the LcYTHDFB subgroup, there is an amino acid substitution in Helix α1, where aspartic acid (at position 401 in human protein) is replaced by asparagine (Figure 4E). Furthermore, the N-termini of the LcYTHDF proteins possess low-complexity regions containing Y/P/Q-rich regions (Figure 4F). However, in LcYTHDCs, the Y/P/Q-rich regions were located between the zinc finger repeat (YTH1 superfamily domain) and the YTH domain (Figure 4F).

### 2.5. LcYTHDF1, LcYTHDF2, LcYTHDF4, LcYTHDF5, LcYTHDF7, and LcCPSF30-L May Participate in the Process of Phase Separation

Considering that human m^6^A reader proteins YTHDF1–YTHDF3 can undergo LLPS in cells with the process being enhanced by multivalent m^6^A modifications, we examined whether LcYTH proteins have liquid-like properties Proteins containing prion-like domains (PrLDs) have been observed to drive proteins to undergo phase separation. In our analysis, we found that *LcYTHDF1*, *LcYTHDF2*, *LcYTHDF4*, *LcYTHDF5*, *LcYTHDF7*, and *LcCPSF30-L* are predicted to contain one or two highly disordered PrLDs (Figure 5). This suggests that these LcYTH proteins may undergo phase separation in a similar manner to the human YTHDF1–YTHDF3 proteins, further supporting their liquid-like properties.

### 2.6. Expression of LcYTHs in Different Tissues

To determine the physiological functions of different members of LcYTHs, we investigated the expression of YTH genes in the leaves, roots, stems, and terminal buds (Figure 6). We found that the LcYTH gene family members were expressed in almost all *L. chinense* (Figure 6). The expression of LcYTHDF1 was slightly higher in leaves, while the expression of LcYTHDF6 was slightly lower in leaves. The expression of LcYTHDF5 and LcYTHDF8 was relatively stable in the four tissues. The expression patterns of LcYTHDF2, LcYTHDF3, LcYTHDF4, and LcYTHDF7 showed similar expression patterns with higher expression in the stems and roots compared to other tissues. The expression of LcCPSF30-L was dominant in stems and inferior in terminal buds (Figure 6A). It is worth mentioning that the expression level of YTHDF2 in the four tissues is much higher than that of other LcYTH genes (Figure 6B).

### 2.7. Analysis of Cis-Acting Elements in LcYTH Promoters

To further investigate which process could be regulated through the LcYTH, the cis-acting elements present in the promoter were analyzed by the online tool PlantCARE. In total, 36 different types of cis-acting elements that could be involved in different processes were identified (Figure 7A). We performed statistics on several major cis-acting components associated with light, MeJA, ABA, drought, low-temperature, GA, auxin, SA, and circadian control (Figure 7B). Among these elements, we observed a high number of ABA and MeJA response elements, with 21 and 26, respectively (Figure 7B; Appendix A). This suggests that the LcYTH genes may be involved in the response to these stresses and hormones (Figure 7B).

### 2.8. Differential Expression Profiling of LcYTH Genes Treated by ABA and MeJA

Studies in plant functional genomics and structural genomics have shown that some *YTH* genes respond to a number of hormones and stresses. In this study, over 77.8% of the LcYTH gene promoter region contained cis-responsive elements for ABA and MeJA, and most of these genes showed inducible expression under hormone treatments (Figure 8). Specifically, under ABA treatment, the relative expression of LcYTHDF1 initially increased and then decreased, while the expression levels of other genes showed a pattern of initial decrease followed by an increase (Figure 8A). In the case of MeJA treatment, apart from LcYTHDF6, eight other members of the LcYTH gene family responded to MeJA treatment (Figure 8B). These eight members exhibited four different expression patterns. In the first category, the expressions of LcYTHDF1, LcYTHDF3, LcYTHDF7, and LcCPSF30-L decreased at about 6–12 h, followed by an increase at 12 h. In the second category, the expression levels of LcYTHDF2 and LcYTHDF4 initially decreased at 6 h, but then increased at 12 h, reaching their highest levels at 24 h before decreasing again. In the third category, the expression of LcYTHDF5 initially increased after MeJA induction, then began to decrease after 6 h, and increased again after 12 h. In the fourth category, the expression of LcYTHDF8 was significantly induced by MeJA treatment.

## 3. Discussion

*N^6^*-methyladenosine, as a research hotspot in recent years, provides a new perspective for biological research. The reader proteins that recognize m^6^A sites play a crucial role in multiple functions of m^6^A modification [1,2]. These m^6^A readers determine the fate of their target RNA, thereby influencing various biological processes [17,18,19]. Therefore, studying m^6^A readers can serve as a starting point to understand how the modification of m^6^A affects specific organs or physiological processes. In plants, YTH proteins have been identified as specific readers that bind to m^6^A in *A. thaliana* [20]. However, the *YTH* gene family has not been identified in *L. chinense*. In this study, we identified the *YTH* gene family and conducted a preliminary analysis of its structural characteristics and expression patterns in *L. chinense*, which laid a foundation for further exploration of the biological functions of YTH in tree growth and development, as well as a response to stress.

### 3.1. The Function of YTH Protein in L. chinense May Be More Complex or Redundant Than That in Animals

In previous studies, Scutenaire identified the *YTH* gene family in *A. thaliana* and a variety of plants [19]. It was observed that the number of *YTH* genes in these plants was higher compared to mammals, which only have five *YTH* genes. This suggests that there may be more YTH protein diversity in plants compared to mammals. In this study, nine YTH proteins were identified from the genome database of *L. chinense*, including two DFA subgroups, two DFB subgroups, four DFC subgroups, and one DCA subgroup (Table 1; Figure 1A–C). Each YTH protein was located on different chromosomes (Figure 2). The higher number of YTH proteins in *L. chinense* compared to mammals is consistent with previous findings. This implies that the regulatory mechanism of YTH protein in plants may be more complex, or there may be serious functional redundancy among members of the YTH family in plants. Functional redundancy among YTH family members has been observed in *A. thaliana*: *ECT2*, *ECT3*, and *ECT4* have functional redundancy in regulating leaf development via the m^6^A–YTH module [24,25]. The results of collinearity analysis and Ka/Ks analysis also support the presence of functional redundancy among the YTH family members (Figure 3).

### 3.2. The LcYTHDC Proteins May Have a Similar but Unique Molecular Mechanism to the YTHDC1 Protein in Animals

Previous research has identified a conserved mechanism utilized by the YTH domain proteins YTHDF and YTHDC to recognize m^6^A [26]. This recognition is facilitated by an aromatic cage that specifically interacts with the methyl moiety of m^6^A. In humans, the YTHDF proteins possess a WWW cage, while the YTHDCs have a WWL-type cage. In this study, we found that the aromatic cage presented in LcYTHs is composed of WWW residues (Figure 4B,E), suggesting that the m^6^A binding mechanism in plant YTHDC proteins may differ from that in animal YTHDC proteins. In mammals, two types of YTHDCs were identified, YTHDC1 and YTHDC2. YTHDC1 contains the YTH domain in the central part of the protein, while YTHDC2 has the YTH domain located at the C-terminus. Interestingly, in *L. chinense*, all YTH domains are located in the central part of the protein, similar to YTHDC1s in mammals (Figure 4F). This suggests that YTHDC proteins in plants may have a similar but unique molecular mechanism compared to YTHDC1 proteins in animals. In mammals, YTHDC1 is known to be involved in various vital processes such as embryo development, cell differentiation, and disease via recognizing m^6^A modification of RNA to mediate gene expression silencing, alternative splicing, nuclear export, RNA stability, and phase separation [14,27,28]. Therefore, it would be interesting to investigate whether LcYTH proteins are also involved in these processes. In the LcYTHDFB subgroup, there is an amino acid substitution in Helix α1, where aspartic acid (at position 401 in human protein) is replaced by asparagine (Figure 4E). This amino acid substitution may result in LcYTHDFB having a higher affinity for m^6^A compared to proteins in the other two subgroups [21]. In most mammal YTHDF proteins, there is a low complexity region in the N-terminal that contains Y/P/Q-rich modules. The Y/P/Q-rich region is thought to regulate the stability of m^6^A-modified mRNA by recruiting proteins containing YTH domains to RNA decay sites [10,29,30]. In this study, Y/P/Q-rich modules were also found in LcYTHs (Figure 4F). Additionally, human YTHDC2, a nucleoplasmic protein, not only contains a C-terminal YTH domain but also contains a DEAD-box RNA helicase domain, which plays a role in unwinding the 5′-UTR of double-stranded RNA, enhancing the translation efficiency of target genes [31,32]. However, YTH proteins containing the DEAD-box RNA helicase domain have not been found in the plant kingdom. Further investigation is required to determine if LcYTH proteins in plants have similar functions and molecular mechanisms as their animal counterparts.

### 3.3. LcYTH Proteins May Participate in LLPS Process through PrLDs Domain

Prion-like domains can induce phase transitions, and further hardening of the droplets leads to pathological fibrous aggregation [33,34]. In mammals, prion-like domains are found in the structure of m^6^A regulatory proteins, indicating that m^6^A-modified RNA is usually associated with phase separation [35,36,37,38]. This phenomenon was also found in the study of Arabidopsis m^6^A readers. In this study, prion-like domains were found in most of the LcYTH proteins, suggesting that these *YTH* genes have the potential to undergo phase separation (Figure 5). This has also been observed in the study of Arabidopsis m^6^A reader proteins.

### 3.4. LcYTHDF2 Has an Obvious Expression Advantage in the LcYTH Gene Family

Understanding gene expression patterns in tissues is crucial for mining functional genes. Previous studies have shown that *YTH* genes were mainly expressed in diverse tissues and were essential for plant growth and development [23]. In our study, we found that LcYTH gene family members are expressed in almost all tissues, but different genes show preferences for expression in specific tissues (Figure 6A). This variation in tissue expression may contribute to the functional differentiation of these genes. It is worth mentioning that the expression level of LcYTHDF2 in these four tissues was significantly higher than that of other *LcYTH* genes (Figure 6B), and a similar phenomenon was found in Arabidopsis [23]. This suggests that LcYTHDF2 may have a unique role in RNA modification.

### 3.5. The LcYTH Genes May Play Important Roles in ABA and MeJA Pathways through Different Forms

YTH family genes have been found to play important roles in hormone and stress responses. To investigate whether LcYTH is involved in stress and hormone regulatory pathways, we evaluated the predicted cis-regulatory elements in all the putative promoter regions of YTH (2000 bp). We found that the promoter region of the LcYTH gene mainly contains developmental and adaptive elements related to plant hormonal and abiotic stresses (Figure 7A). Furthermore, a statistical analysis of these elements revealed a higher abundance of responsive elements for ABA and MeJA (Figure 7B). Suggesting that the YTH family genes might be involved in the process of these responses. In fact, numerous previous studies have demonstrated the involvement of m^6^A regulatory proteins in the ABA regulatory pathway. For example, *AtCPSF30-L* regulates the selective polyadenylation of RPN10 and FYVE1 in the ABA response [17]. *ECT2/3/4* collaboratively promotes the stability of targeted genes by binding PAB protein, thus regulating the molecular mechanism of the ABA response [24]. *ALKBH9B* also regulates the ABA response by regulating mRNA m^6^A levels of ABI1 and BES1, two negative regulators of the ABA signal [39]. In addition, ALKBH10B has been found to be responsive to salt and osmotic stress and induced by two stress response hormones, ABA and JA. Similar expression patterns were also observed in tomatoes and apples [40]. However, there are few reports about the relationship between m^6^A readers and MeJA. To confirm our hypothesis, we studied the expression patterns of YTH family genes in ABA and MeJA treatments. Short-term processing results showed that different LcYTH genes displayed different response patterns to ABA and MeJA, suggesting that LcYTH genes may be involved in the signaling pathways of ABA and MeJA in different forms (Figure 8).

In conclusion, the LcYTH genes play important roles in controlling plant physiology and morphology, particularly under stress conditions. The genome-wide identification of LcYTH gene family members lays an important foundation for further study of the function of these genes. Furthermore, this identification provides a theoretical framework and direction for the application of tree RNA epigenetic molecular breeding.

## 4. Materials and Methods

### 4.1. Identification of YTH Genes in the L. chinense Genome

The consensus YTH RNA-binding domain (containing 106 amino acid residues) sequence was used to identify homologous genes in the *L. chinense* genome database (version 3.1) [41]. We also conducted further BLAST searches in the protein database using an E-value cut-off of 1 × 10^−5^. Based on the YTH domain (PF04146), the hidden Markov model file was downloaded from the Pfam database (http://pfam.xfam.org (accessed on 5 January 2023)). HMMER 3.3.2 (http://hmmer.janelia.org/ (accessed on 5 January 2023)) was selected to search for *YTH* genes in the databases of Iso-Seq. The default parameters were used for screening, and the E-value was set to E < 10^−3^. Pfam (http://pfam.xfam.org/ (accessed on 5 January 2023)) and CD-search (https://www.ncbi.nlm.nih.gov/cdd/ (accessed on 6 January 2023)) were used to screen out protein sequences of *L. chinense* with the YTH domain. Finally, sequences with complete YTH domains were selected and sequences with more than 97% similarity between different databases were deleted.

### 4.2. Amino Acid Sequence Analysis, Multiple Sequence Alignments, and Phylogenetic Analysis

The molecular masses and isoelectric points of the LcYTH proteins were predicted using the web tool ExPASy (https://web.expasy.org/compute_pi/ (accessed on 6 January 2023)). The domains of YTH proteins were analyzed using the web tools CDD (https://www.ncbi.nlm.nih.gov/ (accessed on 6 January 2023)) and ExPASy (https://prosite.expasy.org/ (accessed on 6 January 2023)). Finally, the domain graphs were visualized using TBtools [42]. The motif compositions of LcYTHs were identified by MEME (http://memesuite.org/tools/meme (accessed on 4 March 2023)) and exhibited using TBtools software. Multiple protein sequence alignments were performed using DNAMAN. Statistical support for the major clusters was obtained using aLRT. Bias in amino acid composition was detected using the Composition Profiler website (http://www.cprofiler.org/ (accessed on 4 March 2023)). 

BLAST searches (BLASTP and tblastn) were performed starting from known *A. thaliana* YTH domains on 32 species representing the diversity of the Viridiplantae lineage at the JGI Phytozome (V11) genomic resource (https://phytozome.jgi.doe.gov/pz/portal.html (accessed on 4 March 2023)). To adopt a systematic nomenclature, we decided to name each YTH-containing protein by the initials of its species of origin, followed by either DF or DC, by a number, and possibly by a letter (A, B, or C) referring to a subgroup when appropriate. When another name already existed in the literature, it was kept and its systematic nomenclature name was added in parentheses (see Appendix A for all YTH sequences and for the list of all name codes used in this study). ClustalX2.1 was used for multiple sequence alignments with default parameters. A maximum likelihood phylogenetic tree was constructed using MEGAX (https://www.megasoftware.net/ (accessed on 6 March 2023)) with 1000 bootstrap replicates. The phylogenetic tree was visualized by FigTree (http://tree.bio.ed.ac.uk/software/ (accessed on 7 March 2023)).

### 4.3. Chromosomal Distribution of YTH Genes

The chromosomal location information of the YTH gene family was extracted using the annotation files of the *L. chinense* genomes, and the length of each chromosome was obtained from the NCBI website7; the gene positions were drafted to chromosomes by using MapChart 2.2. To further analyze gene duplication events, the gene repetitive events were analyzed by the Multicollinearity ScanToolkit (MCScanX), and the Advanced Circos function of the TBtools software was used to visualize segmental duplication relationships.

### 4.4. Gene Structure, Conserved Motifs Analysis, and Cis-Regulatory Elements

TBtools software [42] and the online analytical tool KnetMiner (https://knetminer.rothamsted.ac.uk/Triticum_aestivum/ (accessed on 14 March 2023)) were used to construct the map of the exon-intron structure. The conserved motifs of LcYTH proteins were analyzed using Multiple Expectation Maximization for Motif Elicitation (MEME: http://meme-suite.org/tools/meme (accessed on 4 March 2023)) with the following parameters: the minimum and maximum motif widths were 6 and 50, respectively; and the maximum number of motifs was 10. Plant CARE (http://bioinformatics.psb.ugent.be/webtools/plantcare/html/ (accessed on 15 December 2022)) was used to identify cis-regulatory elements in the 2 kb upstream sequences of each expansion gene. The cis-acting elements were visualized using TBtools.

### 4.5. LLPS Prediction

Predictions of PrLDs and disordered regions were made by the prion-like amino acid composition (PLAAC; http://plaac.wi.mit.edu/ (accessed on 6 March 2023)). The predicted results are visualized using AI.

### 4.6. Plant Materials and Treatments

One-year-old *L. chinense* seedlings, grown by the Key Laboratory of Forest Tree Genetic Breeding at Nanjing Forestry University, were used for studying the expression level of LcYTHs subjected to 2 treatments. After 100 μM ABA and 50 μM MeJA were sprayed on plants, leaf samples were collected at 6 h, 12 h, 24 h, and 48 h, then immediately frozen in liquid nitrogen and stored at −80 °C. Next, 3 uniformly growing seedlings were selected for each treatment as 3 biological replicates. Samples collected without any treatment were used as controls.

### 4.7. RNA Extraction and qRT-PCR Analysis

RNA was extracted from the leaves of *L. chinense* using an RNA extraction kit (Vazyme, Nanjing, China). The extracted RNA was then analyzed using 1% agarose gel electrophoresis to confirm its quality and integrity. The RNA was reverse transcribed to synthesize the first strand using a 1-step kit (Vazyme, Nanjing, China). The cDNA was diluted at a concentration of 1:10. Primers were designed based on the sequences of *L. chinense* in the CDS database using Primer 5 software (Appendix A). The Actin97 was used as a reference control gene. Hieff UNICON^®^ Universal Blue qPCR SYBR Green Master Mix (Yeasen, Shanghai, China) was used to detect the target sequences. Each PCR mixture (10 μL) contained 1 μL of diluted cDNA (20 dilution), 5 μL of SYBR Green Real-time PCR Master Mix, 0.4 μL of each primer (10 μM), and 3.2 μL of ddH_2_O. The qRT-PCR reaction was carried out under the following conditions; 1 cycle at 98 °C for 3 min, then 40 cycles at 95 °C for 15 s, 60 °C for 30 s, and 72 °C for 30 s. The 2^−∆∆Ct^ method was used to evaluate gene expression levels [43]. Each sample was analyzed as 3 biological replicates with 3 technical replicates.

### 4.8. Statistical Analysis

For statistical analysis, GraphPad Prism v8.0.2 software was used. All experimental data were obtained from at least 3 replicates, and statistical analysis was performed with a Student’s *t*-test. In all experiments, significant differences in the data were evaluated by a one-way ANOVA. * *p* < 0.05, and ** *p* < 0.01. The gene expression in untreated samples was used as a control for significant analysis.

## 5. Conclusions

In this study, nine *LcYTH* genes were identified from *L. chinense* and phylogenetic analysis and showed that *LcYTHs* can be divided into two classes, LcYTHDF and LcYTHDC. Domain sequence analysis showed that LcYTH proteins had high sequence conservation. The aromatic cage bag of LcYTH is composed of tryptophan and tryptophan (WWW). In addition, PrLDs were found in the protein results of YTH, suggesting that these genes may be involved in the process of liquid–liquid phase separation. Expression analysis of the *LcYTH* genes demonstrated their presence in various tissues and organs, although different YTH genes showed distinct expression preferences across tissues. Moreover, we found that the expression of *LcYTH* genes was responsive to treatments with ABA and MeJA. These findings provide a foundation for future functional analysis of *LcYTH* genes.

## Figures and Tables

**Figure 1 ijms-24-15189-f001:**
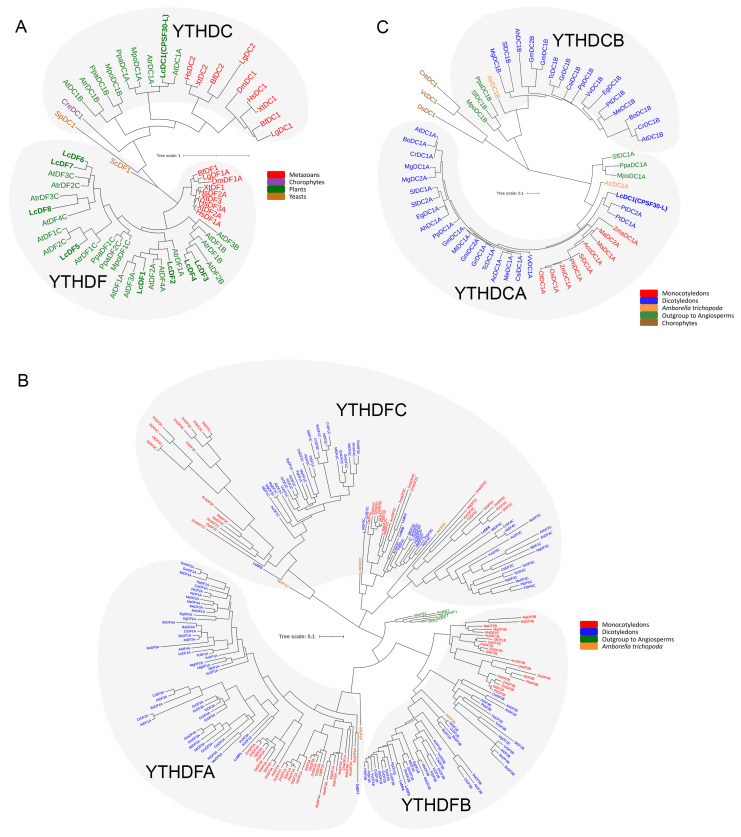
Phylogenetic relationships among different YTH domains from various species including Viridiplantae, metazoan, and yeast core. (**A**) The phylogenetic tree of representative YTH motifs from Viridiplantae compared with several yeast and metazoan YTHs from *Homo sapiens*, *Drosophila melanogaster*, the yeast *Saccharomyces cerevisiae*, and the fission yeast *Schizosaccharomyces pombe* (see Appendix A for YTH sequences). The color code represents different species: brown for yeast, green for plants, blue for chlorophyte species, and red for metazoan species. The scale bar indicates a length of 1 substitution per site; (**B**) The phylogenetic tree of 240 YTHDF proteins from 29 species represents the diversity of the Viridiplantae lineage (see Appendix A for YTH sequences). The color code represents different species: green for outgroup species to angiosperms, orange for *Amborella trichopoda* (*A. trichopoda*), blue for dicotyledon species, and dark red for monocotyledon species. The scale bar indicates a length of 0.1 substitutions per site; (**C**) The phylogenetic tree of 57 YTHDC proteins from 32 species represents the diversity of the Viridiplantae lineage (see Appendix A for YTH sequences. The color code represents different species: brown for chlorophyte species, green for outgroup species to angiosperms, yellow for *A. trichopoda*, blue for dicotyledon species, and dark red for monocotyledon species. The scale bar indicates a length of 0.1 substitutions per site.

**Figure 2 ijms-24-15189-f002:**
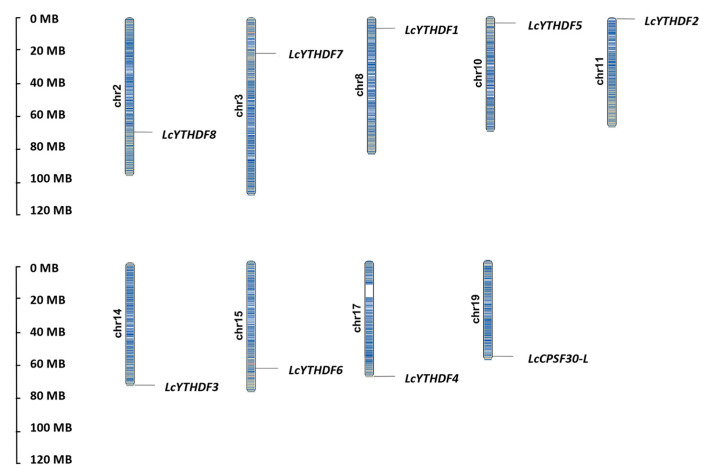
Chromosomal locations of the identified *LcYTHs* in *L. chinense*. The size of a chromosome is expressed by its relative length. The scale bar on the left indicates the chromosome lengths (Mb).

**Figure 3 ijms-24-15189-f003:**
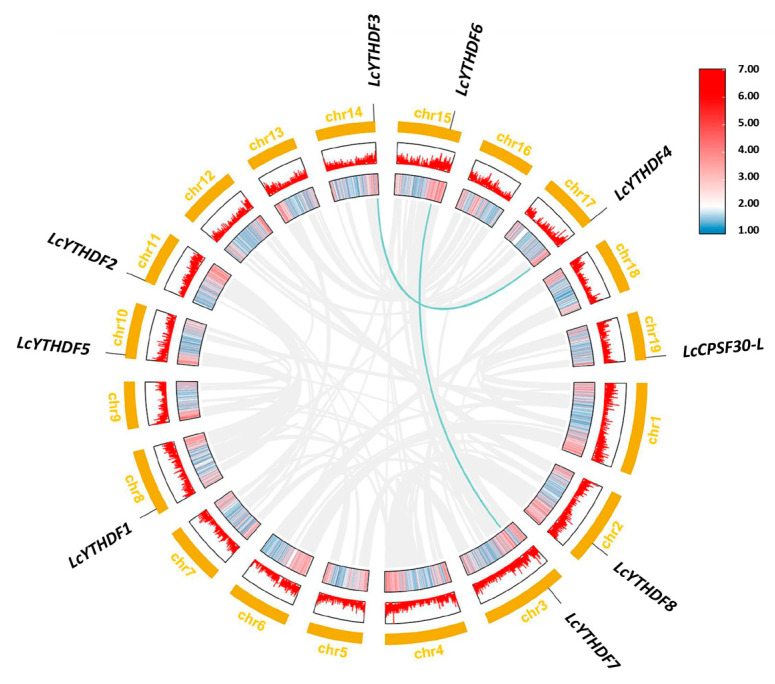
Collinearity mapping of *LcYTH* genes in the *L. chinense* genome. The green lines show homologous gene pairs. From the outside to the inside, the outermost circle represents chromosome coordinates, and the second and third circles represent gene density distribution.

**Figure 4 ijms-24-15189-f004:**
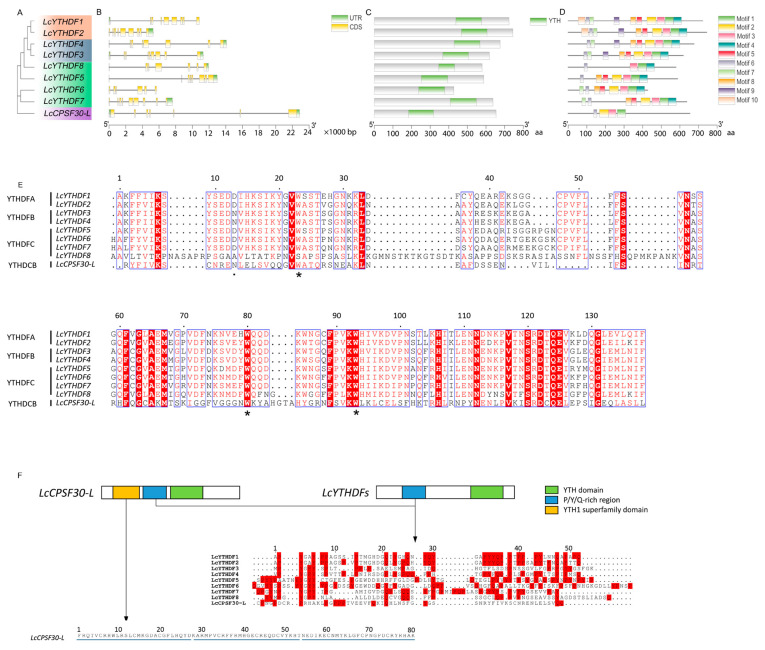
Phylogenetic relationships, domains, exon–intron, and motif structures of LcYTHs. (**A**) The phylogenetic tree was constructed based on the LcYTHs sequences. Nine LcYTHs were clustered into DFA (orange), DFB (silver), DFC (green), and DC (purple) groups; (**B**) Gene structures of the LcYTHs, gray line segments represent introns; (**C**) Conserved domains in LcYTHs; (**D**) The 10 motifs of LcYTHs are distinguished by different colors. (**E**) Sequence alignments of YTH domain in LcYTH family proteins. Asterisks indicate the tryptophan position; The dot indicates that the LcYTHDFB subgroup aspartic acid (located at position 401 in human proteins) is replaced by asparagine. (**F**) Analysis of *LcYTH* genes structure and amino acid sequence. The Y/P/Q amino acids are highlighted in red. Zinc finger repeats are indicated by the blue line.

**Figure 5 ijms-24-15189-f005:**
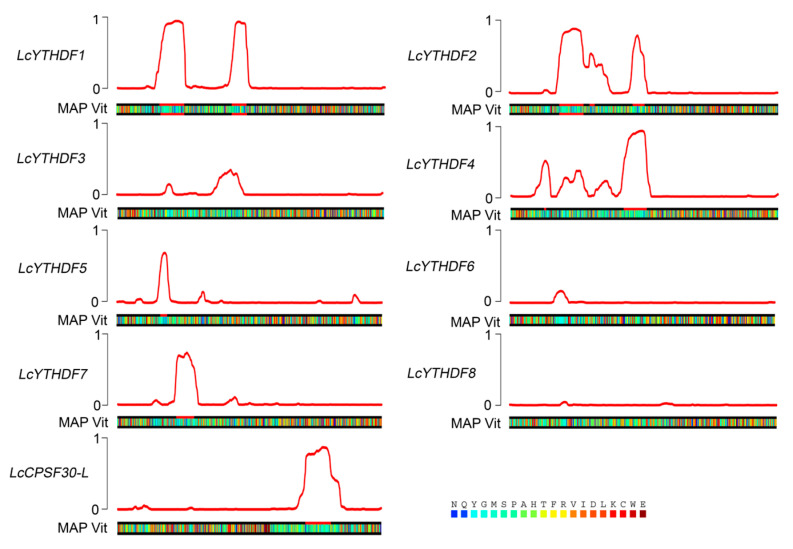
Predictions of PrLDs and disordered regions made by the “prion-like amino acid composition” (PLAAC; http://plaac.wi.mit.edu/ (accessed on 6 March 2023)).

**Figure 6 ijms-24-15189-f006:**
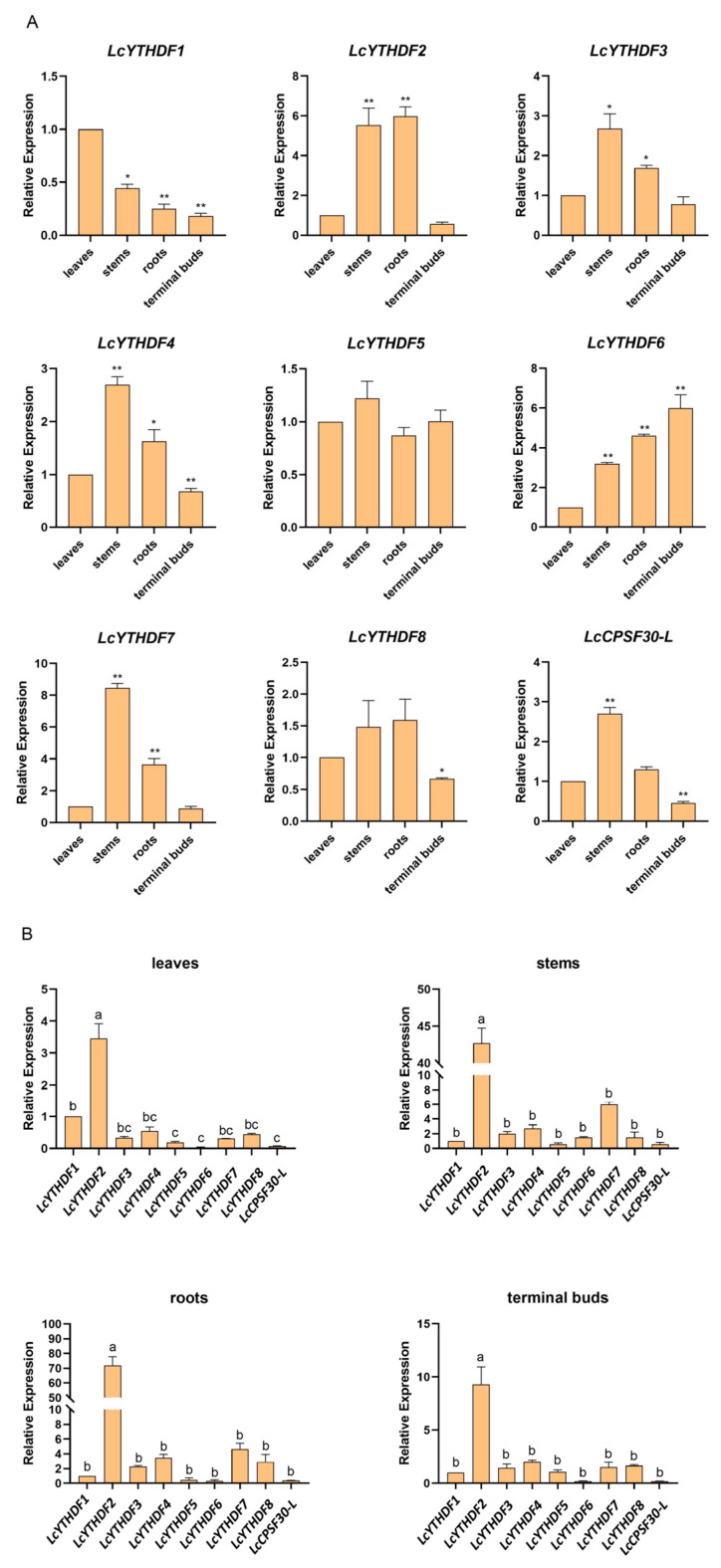
Tissue expression specificity analysis of 9 LcYTHs. (**A**) Expression of 9 LcYTHs in different tissues of *L. chinense*. Relative gene expression was measured using qPCR with *Actin97* as a reference gene, prior to normalization to the expression levels in leaves. Data are shown as mean ± SE, with 3 biological replicates in the experiment. ** *p* < 0.01, * *p* < 0.05, Student’s *t*-test. (**B**) Expression pattern of 9 *LcYTH* genes in the same tissue. The relative expression level was determined using the expression of *LcYTHF1* as a control. Different letters indicate significant differences by using a one-way ANOVA followed by Duncan’s multiple range test.

**Figure 7 ijms-24-15189-f007:**
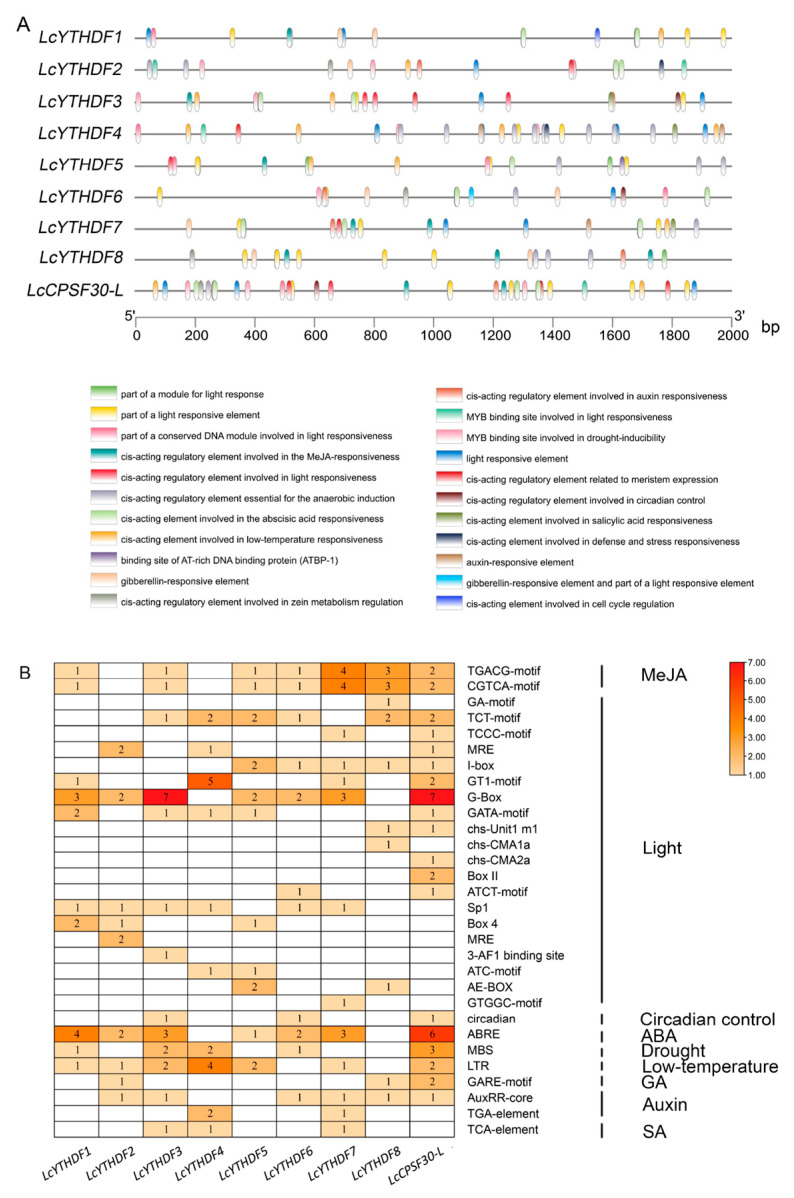
Element analysis of promoter cis-acting elements (**A**) 2000 bp upstream sequence was used to predict the cis-elements. Each of the 9 predicted cis-elements is represented by a different colored box; (**B**) Number of major cis-acting elements of 9 *L. chinense* genes. The blank box indicates that the quantity was 0. Different colors indicate different numbers of cis-acting elements.

**Figure 8 ijms-24-15189-f008:**
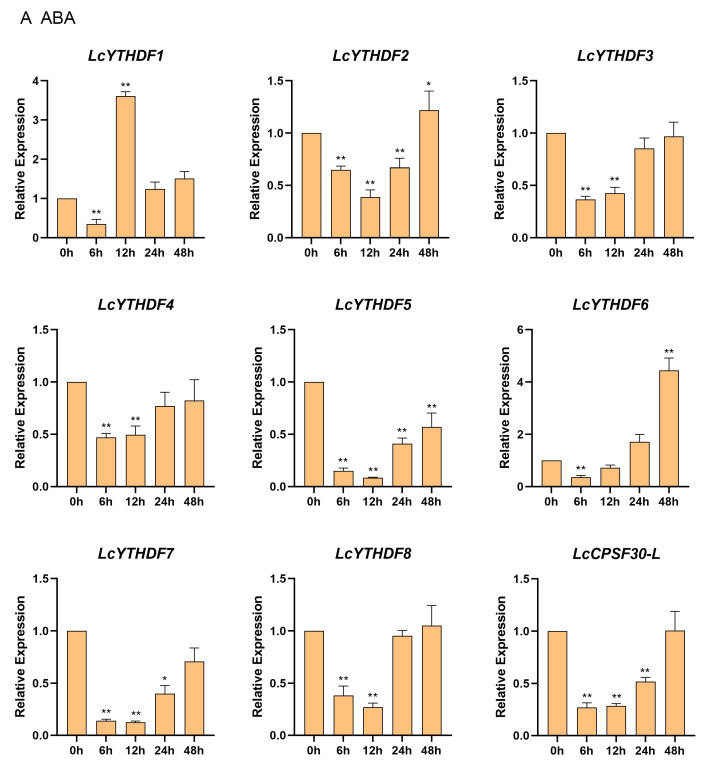
qRT-PCR of expression patterns of *LcYTH* genes under two treatments including ABA (**A**) and MeJA (**B**). The relative expression level in untreated samples was used as the control for significant analysis. Means of triplicates ± SE, ** *p* < 0.01, * *p* < 0.05, Student’s *t*-test.

**Table 1 ijms-24-15189-t001:** Summary of data relating to the 9 LcYTHs proteins.

Group	Gene ID	Locus ID	Locus	Protein Length (aa)	MW (kDa)	pI	Homolog in *A. thaliana*
DFA	*LcYTHDF1*	Lchi30795	Chr08	723	78.3	6.04	*ECT1/2/3/4*
DFA	*LcYTHDF2*	Lchi30011	Chr11	744	80.4	6.56	*ECT1/2/3/4*
DFB	*LcYTHDF3*	Lchi10804	Chr14	618	67.2	5.43	*ECT5/9/10*
DFB	*LcYTHDF4*	Lchi05236	Chr17	679	73.3	5.31	*ECT5/9/10*
DFC	*LcYTHDF5*	Lchi06362	Chr10	588	64.4	6.25	*ECT6/7*
DFC	*LcYTHDF6*	Lchi04058	Chr15	427	48.2	5.97	*ECT8*
DFC	*LcYTHDF7*	Lchi07049	Chr03	637	70.1	6.07	*ECT8*
DFC	*LcYTHDF8*	Lchi01573	Chr04	579	61.9	6.44	*ECT11*
DCA	*LcCPSF30-L* (*LcDC1*)	Lchi11130	Chr19	654	71.8	5.81	*CPSF30-L*

## Data Availability

Not applicable.

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
