# Peer review of "Genome-Wide Identification and Characterization of the YTH Domain-Containing RNA-Binding Protein Family in Liriodendron chinense"

_ijms, 2023, doi:10.3390/ijms242015189_

Round 1

Reviewer 1 Report

In the MS ijms-2613777, the authors claim the reader's interest with a detailed molecular study on Liriodendron chinense, a rare and precious tree from Chinese woods. The authors identified the LcYTH gene in L. chinense. They investigated its expression pattern in different tissues and its response to phytohormones and abiotic stress. The present study aims to show a theoretical framework for gaining deeper insights into the molecular mechanisms and functions of YTH genes in the growth, development, and stress resistance of L. chinense. 

The MS is well organized. From 46 references of the current version, 25 have been published in the last 5 years.

The following comments and suggestions are available below:

1. Lines 96-100. Please reformulate the big phrase, separating it into two fragments to avoid repetition and better understanding.

2. Figure 1. Please augment the smallest letters or increase the resolution for better visualization. Approximately all colored letters and those from the legend are too small and blurred. The reviewer tried to zoom the figure but without better effects.

3. Similar comments are available for Figures 3, 4, 5, 7. Figure 2 is more clear, but increasing its size is necessary.

4. Figures 6 and 8. Please explain the significance of *, **, and small letters above the columns for better understanding.

5. Lines 246, 283. The authors stated: "Error bars represent means ± SE from three independent experiments." It is SE or Standard Deviation (SD)?

6. Lines 104, 105, 106, 107, 146, 436,  L. chinense and other species scientific names with italics. Please check again and correct all misprints in the entire MS.

7. Line 439. "leaf samples were collected at 0 h (without any treatment)" - maybe without 0 h is better. The same suggestion is available for line 441 - it is good enough "untreated sample" for example. Please, check, correct, and revise the control presentation in the entire MS. For example:

- Line 461-462. "The gene expression during 0 h stress treatment was used as a control for significant analysis."

- Lines 245-246. "The relative expression level was determined using the expression of in leaves as a control"

- Lines 283-284."The relative expression level was determined with 0 h of treatment as the control."

Author Response

Point-by-point responses

Reviewer #1:
The following comments and suggestions are available below:

  1. Lines 96-100. Please reformulate the big phrase, separating it into two fragments to avoid repetition and better understanding.

Response #1: We are sorry for these mistakes. Based on the suggestion, we have now reformulated this big phrase in line 96-100. We would like to take this opportunity again to thank the reviewer for the very helpful comments that dramatically improve our manuscript.

  1. Figure 1. Please augment the smallest letters or increase the resolution for better visualization. Approximately all colored letters and those from the legend are too small and blurred. The reviewer tried to zoom the figure but without better effects.

Response #2: Thanks for the suggestion. We have optimized the resolution and the letter size in Figure 1. We do appreciate these improvements in Figure 1 which gives the readers a better visualization.

  1. Similar comments are available for Figures 3, 4, 5, 7. Figure 2 is more clear, but increasing its size is necessary.

Response #3: Thanks for the suggestion. We have optimized the resolution and the letter size in Figure 3, 4, 5, 7. We do appreciate these improvements in these figures which gives the readers a better visualization.

  1. Figures 6 and 8. Please explain the significance of *, **, and small letters above the columns for better understanding.

Response #4: We thank this referee for the very positive comments and thorough reviews. All the suggestions provided by this referee before are great for us. We have added “Means of triplicates ± SE, ** p < 0.01, * p < 0.05, Student’s t-test.” to explain the significance of *, ** in Figure 6, and “Different letters indicate significant differences by using one-way ANOVA followed by the Duncan’s multiple range test.” to explain the significance of small letters above the columns in Figure 8.

  1. Lines 246, 283. The authors stated: "Error bars represent means ± SE from three independent experiments." It is SE or Standard Deviation (SD)?

Response #5: We are sorry for these mistakes. Based on these suggestions, we have now corrected “Error bars represent means ± SE from three independent experiments” into “Data are shown as mean ± SE, with three biological replicates in the experiment” in line 246, 283. We would like to take this opportunity again to thank the reviewer for the very helpful comments that dramatically improve our manuscript.

  1. Lines 104, 105, 106, 107, 146, 436, chinense and other species scientific names with italics. Please check again and correct all misprints in the entire MS.

Response #6: We thank this referee for the very positive comments and thorough reviews. All the suggestions provided by this referee before are great for us. We have checked again and corrected all misprints of species scientific names in the entire MS.

  1. Line 439. "leaf samples were collected at 0 h (without any treatment)" - maybe without 0 h is better. The same suggestion is available for line 441 - it is good enough "untreated sample" for example. Please, check, correct, and revise the control presentation in the entire MS. For example:

- Line 461-462. "The gene expression during 0 h stress treatment was used as a control for significant analysis."

- Lines 245-246. "The relative expression level was determined using the expression of in leaves as a control"

- Lines 283-284."The relative expression level was determined with 0 h of treatment as the control."

Response #7: We thank this referee for the very positive comments and thorough reviews. All the suggestions provided by this referee before are great for us. We have deleted “0 h (without any treatment)” in line 439, and corrected “Samples collected at 0 h were used as controls” into “Samples collected without any treatment were used as controls” in line 455. Based on the suggestions, we also corrected “The gene expression during 0 h stress treatment was used as a control for significant analysis” into “The gene expression in untreated samples was used as a control for significant analysis” in line 474-475, and “The relative expression level was determined using the expression of in leaves as a control” into “Relative gene expression was measured using qPCR with ACTIN97 as a reference gene, prior to normalization to the expression levels in leaves” in lines 231-232, and “The relative expression level was determined with 0 h of treatment as the control” into “The relative expression level in untreated samples was used as the control for significant analysis” in lines 283-284. All the points raised by this referee are very good suggestions for us to further improve our manuscript.

Reviewer 2 Report

The manuscript is very interesting, well prepared and it contains valuable new data. It can be published after minor revision concerning few language and editorial mistakes.

Line 33. Please change “regulate” to regulates (N6-methyladenosine regulates…)

Line 39. “Therefore, it is necessary to investigate plant m6A modification.” This sentence is awkward, please modify it (e.g., “Therefore, the aim of the present study was to investigate a plant m6A modification system”.

Line 56, Please change “researches” to research (According to the current research).

Line 82. Insert “the” or “this” (Additionally, this research…)

Line 92. Please add: Liriodendron chinense, a tree species of the Magnoliaceae family, an ancient relict plant, … The meaning of the term: “ancient plant” can be wrongly understood without the context.

Lines 103-106, and elsewhere. All Latin names of the organisms, including algae, bacteria etc. should be written in Latin! Please carefully check the text of the manuscript, because there is a lot of such mistakes. Arabidopsis can be written normally, but when you used the full Latin name Arabidopsis thaliana, it should be also written in italic!

Line 124. Table 1. “Summary of information…” Please change to “Summary of data..”

Line 271. Delete additional dot after Figure 8A.

Line 277. Does “thirdly” mean “third category”? If yes, please write consequently “In the third category”.

Line 279. Does “the last category” means the third one? Please clarify this part of the text, this “thirdly” is making a confusion, afterwards “the last category” is not understandable.

Lines 297, 304 etc. again, please check carefully Latin names – and write them in italics.

Author Response

Reviewer #2:
The manuscript is very interesting, well prepared and it contains valuable new data. It can be published after minor revision concerning few language and editorial mistakes.

  1. Line 33. Please change “regulate” to regulates (N6-methyladenosine regulates…)

Response #1: Sorry for the mistake. We have corrected “regulate” into “regulates” in line 36. We would like to take this opportunity again to thank the reviewer for the very helpful comments that dramatically improve our manuscript.

2.Line 39. “Therefore, it is necessary to investigate plant m6A modification.” This sentence is awkward, please modify it (e.g., “Therefore, the aim of the present study was to investigate a plant m6A modification system”.

Response #2: Thanks for the suggestion. We have corrected “Therefore, it is necessary to investigate plant m6A modification” into “Therefore, the aim of the present study was to investigate a plant m6A modification system” in line 43. We would like to take this opportunity again to thank the reviewer for the very helpful comments.

3.Line 56, Please change “researches” to research (According to the current research).

Response #3: Thanks for the suggestion. We have corrected “researches” into “research” in line 60. We would like to take this opportunity again to thank the reviewer for the very helpful comments.

4.Line 87. Insert “the” or “this” (Additionally, this research…)

Response #4: Thanks for the suggestion. We have corrected “Additionally, research has demonstrated that the recognition of m6A sites by CPSF30-L affects the process of liquid-liquid phase separation (LLPS)” into “Additionally, this research has demonstrated that the recognition of m6A sites by CPSF30-L affects the process of liquid-liquid phase separation (LLPS)” in line 87. We would like to take this opportunity again to thank the reviewer for the very helpful comments.

5.Line 92. Please add: Liriodendron chinense, a tree species of the Magnoliaceae family, an ancient relict plant, … The meaning of the term: “ancient plant” can be wrongly understood without the context.

Response #5: Thanks for the suggestion. We have added “a tree species of the Magnoliaceae family, an ancient relict plant” in line 97. We would like to take this opportunity again to thank the reviewer for the very helpful comments.

6.Lines 103-106, and elsewhere. All Latin names of the organisms, including algae, bacteria etc. should be written in Latin! Please carefully check the text of the manuscript, because there is a lot of such mistakes. Arabidopsis can be written normally, but when you used the full Latin name Arabidopsis thaliana, it should be also written in italic!

Response #6: Sorry for the mistakes. We have carefully checked and corrected all Latin names of the organisms in italic in the text of the manuscript. All the points raised by this referee are very good suggestions for us to further improve our manuscript.

7.Line 124. Table 1. “Summary of information…” Please change to “Summary of data..”

Response #7: Thanks for the suggestion. We have corrected “Summary of information” into “Summary of data” in line 115. We would like to take this opportunity again to thank the reviewer for the very helpful comments.

8.Line 271. Delete additional dot after Figure 8A.

Response #8: Thanks for the suggestion. We have Delete additional dot after Figure 8A. in line 270. We thank this referee for the very positive comments and thorough reviews.

9.Line 277. Does “thirdly” mean “third category”? If yes, please write consequently “In the third category”.

Response #9: We are sorry for the mistake. We have corrected “Thirdly” into “In the third category” in line 266. We would like to take this opportunity again to thank the reviewer for the very helpful comments.

10.Line 279. Does “the last category” means the third one? Please clarify this part of the text, this “thirdly” is making a confusion, afterwards “the last category” is not understandable.

Response #10: We are sorry for the mistake. We have corrected “In the last category” into “In the fourth category” in line 268. We would like to take this opportunity again to thank the reviewer for the very helpful comments.

11.Lines 297, 304 etc. again, please check carefully Latin names – and write them in italics.

Response #11: Thanks for the suggestion. We have checked carefully the Latin names and write them in italics in line 297, 304 etc. We would like to take this opportunity again to thank the reviewer for the very helpful comments.

Reviewer 3 Report

Dear Editors of International Journal of Molecular Sciences,

Thank you for choosing me as a reviewer of the of the manuscript ID: ijms-2597039 entitled: ,,Genome-wide identification of members in the YTH domain-containing RNA-binding protein family in Liriodendron chinense and expression analysis of their responsiveness to ABA and MeJAI hope that my comments will help authors to improve their manuscript.

Detailed remarks concerning the manuscript:

I suggest to modify the title of the manuscript to be concise and informative.

Abstract: As it was suggested in the guidelines for Authors I suggest to prepare structured abstracts, but without headings: 1) Background: Place the question addressed in a broad context and highlight the purpose of the study; 2) Methods: Describe briefly the main methods or treatments applied. Include any relevant preregistration numbers, and species and strains of any animals used; 3) Results: Summarize the article's main findings; and 4) Conclusion: Indicate the main conclusions or interpretations.

Key word: It is not recommended to use as key words the words or phrases used in the title of the manuscript. Please do needed changes.

Results section needs improvement. It is too long and wordy. Besisdes some sentences included in this section sounds like discussion: ,,Previous studies have shown that at least one protein version from each DF subgroup 116 in all angiosperm species. We further constructed an evolutionary tree specifically for the 117 DF genes” or „The analysis of LcYTH proteins revealed that the YTH domain, which is responsible 186 for binding the methylysine residue of m6A, is the only recognizable module at their C- 187 terminus (Figure 4C). This observation is consistent with other species [26-27]” or „In the 198 LcYTHDFB subgroup, there is an amino acid substitution in Helix α1, where aspartic acid 199 (at position 401 in human protein) is replaced by asparagine (Figure 4E). This amino acid 200 substitution may result in LcYTHDFB having a higher affinity for m6A compared to pro- 201 teins in the other two subgroups [21]”. The section results should not contain reference citation. Therefore both the Results and Discussion sections should be modified.

I suggest to divide the “Discussion” section into the subsections corresponded to the Results section and discuss each subsection separately. The discussion section should not contain citations of tables or figures. It should be description of the obtained results on the background of the studies of the other Authors. It should be also the interpretation by the Authors of the results presented in the manuscript.

The conclusions should be shortened.

Please indicate the practical application for the study.

References should be prepared strictly to the guidelines for authors. There are many editorial mistakes that should be improved. There are some examples:

a)      Once each word of the  manuscript title is written with capital letter, but the other time only the first word of the manuscript title is written with capital letter.

b)      Once the full, but the other time abbreviated Journal names are provided. See: „Liao, S.; Sun, H.; Xu, C. YTH domain: a family of N6-methyladenosine (m6A) readers. Genomics Proteomics Bioinformatics. 2018, 556 16, 99-107.” and „Fu, Y.; Zhuang, X. m6A-binding YTHDF proteins promote stress granule formation. Nat Chem Biol. 2020, 16, 955-963”

c)      All the Latin names as well generic name should be italicized.

d)     See reference 25. The Journal title should not be bolded.

Please go through the whole reference and do needed changes.

Author Response

Reviewer #3:
Dear Editors of International Journal of Molecular Sciences,

Thank you for choosing me as a reviewer of the of the manuscript ID: ijms-2597039 entitled: ,,Genome-wide identification of members in the YTH domain-containing RNA-binding protein family in Liriodendron chinense and expression analysis of their responsiveness to ABA and MeJA” I hope that my comments will help authors to improve their manuscript.

Detailed remarks concerning the manuscript:

1.I suggest to modify the title of the manuscript to be concise and informative.

Response #1: Thanks for the suggestion. We have modified the title as “Genome-wide identification and characterization of the YTH domain-containing RNA-binding protein family in Liriodendron chinense”. We would like to take this opportunity again to thank the reviewer for the very helpful comments.

2.Abstract: As it was suggested in the guidelines for Authors, I suggest to prepare structured abstracts, but without headings: 1) Background: Place the question addressed in a broad context and highlight the purpose of the study; 2) Methods: Describe briefly the main methods or treatments applied. Include any relevant preregistration numbers, and species and strains of any animals used; 3) Results: Summarize the article's main findings; and 4) Conclusion: Indicate the main conclusions or interpretations.

Response #2: Thanks for the suggestion. Base on the suggestion, we have revised the abstract carefully. We would like to take this opportunity again to thank the reviewer for the very helpful comments.

3.Key word: It is not recommended to use as key words the words or phrases used in the title of the manuscript. Please do needed changes.

Response #3: Thanks for the suggestion. We have corrected and used “RNA methylation; m6A; Liriodendron chinense; YTH domain; expression profiling” as the key words. We would like to take this opportunity again to thank the reviewer for the very helpful comments.

4.Results section needs improvement. It is too long and wordy. Besisdes some sentences included in this section sounds like discussion: ,,Previous studies have shown that at least one protein version from each DF subgroup 116 in all angiosperm species. We further constructed an evolutionary tree specifically for the 117 DF genes” or „The analysis of LcYTH proteins revealed that the YTH domain, which is responsible 186 for binding the methylysine residue of m6A, is the only recognizable module at their C- 187 terminus (Figure 4C). This observation is consistent with other species [26-27]” or „In the 198 LcYTHDFB subgroup, there is an amino acid substitution in Helix α1, where aspartic acid 199 (at position 401 in human protein) is replaced by asparagine (Figure 4E). This amino acid 200 substitution may result in LcYTHDFB having a higher affinity for m6A compared to pro- 201 teins in the other two subgroups [21]”. The section results should not contain reference citation. Therefore both the Results and Discussion sections should be modified.

Response #4: Thanks for the suggestion. We have revised the Results section as suggested above and all the revised places have been marked in blue color front. We would like to take this opportunity again to thank the reviewer for the very helpful comments.

5.I suggest to divide the “Discussion” section into the subsections corresponded to the Results section and discuss each subsection separately. The discussion section should not contain citations of tables or figures. It should be description of the obtained results on the background of the studies of the other Authors. It should be also the interpretation by the Authors of the results presented in the manuscript.

Response #5: Thanks for your professional advice! We will put the “Discussion” section into the subsections corresponded to the “Result” section and mark the modified part in blue font. I believe this will make the article more organized.

6.The conclusions should be shortened.

Response #6: Thanks for the suggestion. The conclusions should be shortened as suggested. We would like to take this opportunity again to thank the reviewer for the very helpful comments.

7.Please indicate the practical application for the study.

Response #7: Thanks for the suggestion. In this study, we identified and analyzed the phylogenetic structure, chromosome distribution, amino acid sequence and expression pattern of 9 LcYTH genes. Our results laid a foundation for further study of the function of LcYTH gene and further genetic and functional analyses of m6A RNA modification in forest trees. We would like to take this opportunity again to thank the reviewer for the very helpful comments.

8.References should be prepared strictly to the guidelines for authors. There are many editorial mistakes that should be improved. There are some examples:

  1. a)      Once each word of the  manuscript title is written with capital letter, but the other time only the first word of the manuscript title is written with capital letter.

Response #8a: Sorry for the mistakes. We have corrected and used the style that only the first word of the manuscript title is written with capital letter in the references. We would like to take this opportunity again to thank the reviewer for the very helpful comments. All the points raised by this referee are very good suggestions for us to further improve our manuscript.

  1. b)      Once the full, but the other time abbreviated Journal names are provided. See: „Liao, S.; Sun, H.; Xu, C. YTH domain: a family of N6-methyladenosine (m6A) readers. Genomics Proteomics Bioinformatics. 2018, 556 16, 99-107.” and „Fu, Y.; Zhuang, X. m6A-binding YTHDF proteins promote stress granule formation. Nat Chem Biol. 2020, 16, 955-963”

Response #8b: Sorry for the mistakes. We have corrected and used abbreviated Journal names in the references. All the corrected places have been marker in blue front. We would like to take this opportunity again to thank the reviewer for the very helpful comments.

  1. c)      All the Latin names as well generic name should be italicized.

Response #8c: Thanks for the suggestion. We have checked and corrected all the Latin names as well generic name in italic. We would like to take this opportunity again to thank the reviewer for the very helpful comments.

  1. d)     See reference 25. The Journal title should not be bolded.

Please go through the whole reference and do needed changes.

Response #8d: Sorry for the mistakes. We have corrected the Journal names in the reference 25. We would like to take this opportunity again to thank the reviewer for the very helpful comments.

Round 2

Reviewer 3 Report

Dear Editors,

Once again thank you so much for choosing me as a reviewer of the manuscript ijms-2613777 entitled: ‘Genome-wide identification of members in the YTH domain-containing RNA-binding protein family in Liriodendron chinense and expression analysis of their responsiveness to ABA and MeJA. I would like to thank the Authors for their efforts in improving their manuscript according to the suggested comments. The report may be accepted.   Therefore there are still some small changes to introduce.   1. Line 20. Liriodendron chinense (L. chinense). Why the abbreviated Latin name of the specvies is used together with its full name? Do needed changes in the whole manuscript. 2. As it was suggested the practical application of the study should be given, but it is still missed. 3.  Typically information concerning the aim of the study is placed at the end of the introduction section. Please do needed changes in order to include the aim of the study together with needed description at the end of the manuscript.

3. As it was claimed it is not suggested to use as key words the words or phrases used in the title of the manuscript. The suggested changes was not introduced. There are still the same words or phrases used in the title of the manuscript and as key words.